# Clonal reconstruction from co-occurrence of vector integration sites accurately quantifies expanding clones in vivo

Sebastian Wagner [1,3], Christoph Baldow [1,3], Andrea Calabria [2], Laura Rudilosso[2], Pierangela Gallina[2], Eugenio Montini [2], Daniela Cesana [2] & Ingmar Glauche [1✉]

High transduction rates of viral vectors in gene therapies (GT) and experimental hemato-poiesis ensure a high frequency of gene delivery, although multiple integration events can occur in the same cell. Therefore, tracing of integration sites (IS) leads to mis-quantification of the true clonal spectrum and limits safety considerations in GT. Hence, we use correlations between repeated measurements of IS abundances to estimate their mutual similarity and identify clusters of co-occurring IS, for which we assume a clonal origin. We evaluate the performance, robustness and specificity of our methodology using clonal simulations. The reconstruction methods, implemented and provided as an R-package, are further applied to experimental clonal mixes and preclinical models of hematopoietic GT. Our results demon-strate that clonal reconstruction from IS data allows to overcome systematic biases in the clonal quantification as an essential prerequisite for the assessment of safety and long-term efficacy of GT involving integrative vectors.

---

[1] Institute for Medical Informatics and Biometry, Carl Gustav Carus Faculty of Medicine, Technische Universität Dresden, Dresden, Germany. [2] San Raffaele-Telethon Institute for Gene Therapy (SR-Tiget), IRCCS Ospedale San Raffaele, Milan, Italy. [3]These authors contributed equally: Sebastian Wagner, Christoph Baldow. ✉email: ingmar.glauche@tu-dresden.de

Gene therapy (GT) approaches aim to compensate the missing functionality of a mutated gene by the insertion of one or more corrected copies of the same gene into the genome of patients' cells. Integrative viral vectors, such as gamma retro viral and lentiviral vectors (LV), are clinically important vehicles to realize the permanent integration of a therapeutic transgene into the genome of hematopoietic stem and progenitors cells (HSPC) to establish the expression of the defective gene in all the cells of the hematopoietic systems[1]. The semi-random integration site (IS) of the target sequence within the host genome represents an inheritable "fingerprint" allowing to track the progeny of the targeted cells over time and in different locations. However, the permanent gene transfer confers the risk of disturbing a cell's genetic program potentially leading to uncontrolled proliferation[2]. Since the first reports about insertional mutagenesis events in early clinical GT trials[3], the continuous monitoring of IS abundances has become a standard method to detect aberrant and potentially malignant clonal expansion[4]. Therefore, corresponding protocols for the assessment of safety and efficacy are routinely implemented in GT trials.

The availability of molecular approaches for IS retrieval[5] utilizing next generation sequencing (NGS) has greatly improved the efficiency to identify and quantify the abundance of multiple IS within one sample[6]. Recent advances in PCR methods allowed a reliable quantification of IS abundance with a detection limit in the order of 0.1%[7]. However, the achievement of a better and reproducible correlation between input material and the number of identified IS is still challenging and relies on many factors such as the overall level of gene marking, the amount of available DNA, or the methodology for IS calling and quantification[8,9]. Nevertheless, it is generally accepted that the IS abundance also corresponds to the clonal abundance, although this interpretation is only valid if each clone is solely marked by a single IS[10]. Aiming towards high transduction rates among initially transplanted cells, target cells commonly integrate more than one virally transduced genetic sequence. The incorporation of multiple IS within individual cells strongly affects the interpretation of quantitative IS analysis. For the example in Fig. 1, we consider three clones with two, four, or six IS. The clonal ground truth in Fig. 1a indicates that one clone is clearly dominating. From the measurement side, only time courses of IS abundance are available (Fig. 1b), while it is a priori not clear whether and how they belong to distinct clones. The example illustrates that naïvely

assuming that each IS represents a unique clone misinterprets the true level of clonality and misses the substantial dominance of one of the clones. Unfortunately, this missing association between IS and clones is a fundamental challenge in GT applications that increases the risk of overestimating the overall clonality and underestimating single clonal abundances. Especially in clinical settings, in which the safety assessment of GT trials is explicitly based on relative IS abundances[11–16], it is essential to know about and correct for multiple integrations.

Related questions about the assignment of mutational variants to the dynamically accumulating clonal structures in genotypically heterogenous tumors have been raised in the field of cancer sequencing and evolution[17–19]. Corresponding reconstruction methods are based on the independent frequency of variant alleles and aim to recover the underlying clonal structure. In the context of gene therapy with viral vectors, one or more integrations may occur in the same cell at the time of transduction while no new integrations appear in vivo after transplantation. IS abundance is only measurable relative to the abundance of all other IS and the reconstruction of clones aims at correlating all observed IS read counts without assuming any hierarchical relation. To our knowledge, no methods are available to address this issue and to correct clone size quantification in the context of gene therapeutic applications. Therefore, we here develop a bioinformatic approach to detect co-occurrent IS within the same clone and provide an R software package *MultIS* for the corresponding analysis. Our approach is based on the idea that two IS of the same clone appear in a constant relative frequency to each other while IS from different clones will change their mutual relative frequency according to the corresponding different clone sizes. We use a mathematical modeling approach to illustrate how the identification of mutual correlations between all pairs of IS can be used to identify sets of IS with high similarity, suggesting the same clonal origin. We particularly employ mathematical modeling to demonstrate both the potential and limitation of this approach. We further validate our method using in vitro data obtained by mixing multiple cell clones with different but known IS. Finally, we apply our pipeline to a mouse model and to available primate data reflecting real case scenarios of HSPC-GT. Hence, we develop and experimentally validate a method that accurately identifies source clones from observed IS, improving downstream analyses for preclinical and clinical studies.

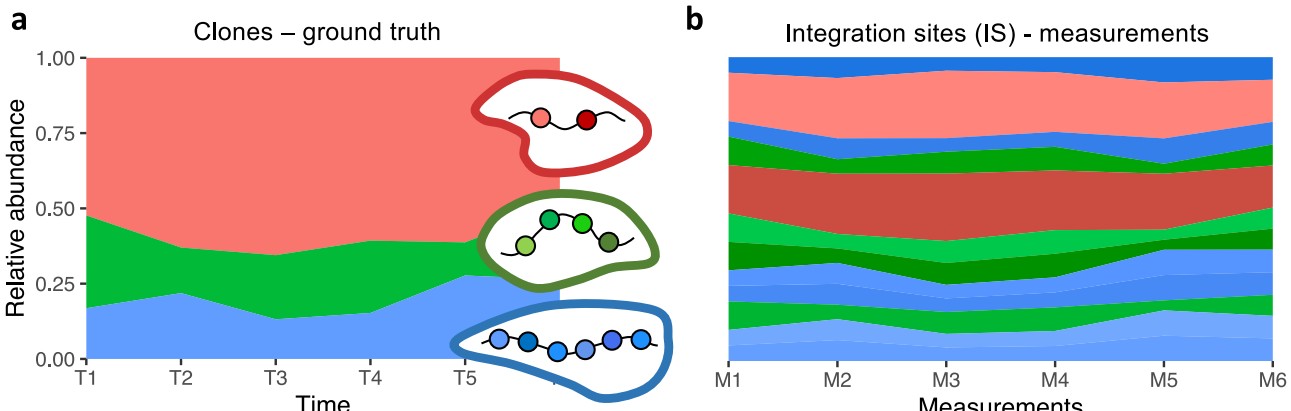

**Fig. 1 Sketched example of a clonal time course and the corresponding IS measurements. a** Artificial example of a time course depicting the relative abundance of three clones (i.e., clonal ground truth). The insets indicate that each clone is identified by a variable number of up to six IS. **b** The corresponding measurement of the IS detects a total of 12 IS. Although the colors indicate the clonal origin, this information is usually not known a priori. Our suggested pipelines aim to use these relative IS abundances to reconstruct the common clonal origin of the different IS and to estimate the true, underlying clonal time course as depicted in **a**.

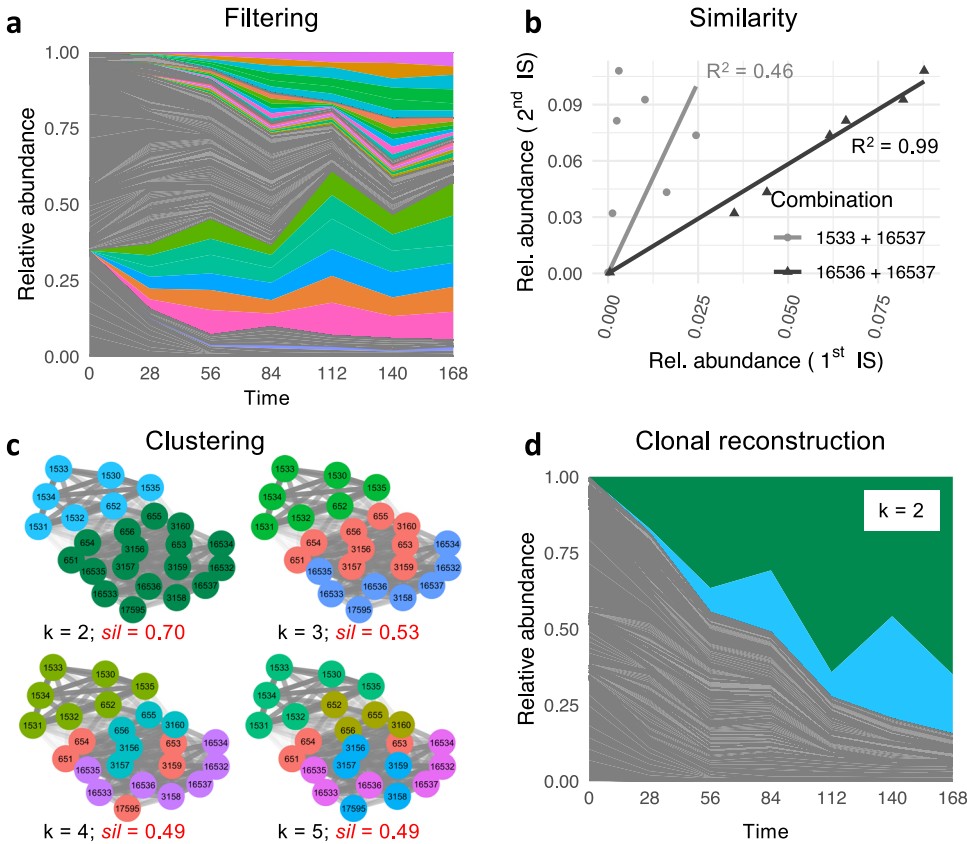

**Fig. 2 Overview of the reconstruction pipeline. a** Example of a time series of IS abundance after applying a filtering step (IS > 1% at the last time point). Gray colors indicate IS that did not pass the filtering. **b** Two examples for the calculation of the similarity score $R^2$ between two pairs of IS. The smaller residuals between triangles and the black line indicate a higher similarity between IS 16536 and 16537 ($R^2 = 0.99$) as compared to IS 1533 and 16537 ($R^2 = 0.46$), which are shown in gray and present with larger residuals. **c** Network representations of the corresponding similarity matrix $S$ in which the shading of the edges indicates the mutual correlation between the IS. We show four different clusterings for increasing values of clusters $k = 2$ to 5, for which the coloring of the nodes represents the obtained clustering. Using the silhouette score *sil* (provided in red) we compare the overall clustering quality between these results, indicating that for the particular data set, $k = 2$ clusters optimally reflect the structure of the similarity matrix $S$. **d** The clonal time series which has been reconstructed by assigning IS of one cluster to the same clone. IS that did not pass the filtering, are still shown in gray.

## Results

**Clonal reconstruction for simulated time-series data**. We present a bioinformatic pipeline to identify IS that belong to the same clone. Our suggested method is based on the idea that multiple IS of the same clone appears with similar relative abundances. If the source clone is small, all its IS should present with low abundance, while IS of a dominant clone should all appear more frequent. Following an initial *filtering* step we calculate pairwise *similarities* among all IS which are fed into a *clustering* algorithm that identifies co-occurring, *clonally* related IS (Fig. 2, Methods and Supplementary Note 1).

In order to demonstrate the general feasibility of our approach and to learn about its limitation, we implemented a scalable mathematical model mimicking time series of clonal development (Methods and Supplementary Note 1). In brief, our model describes the proliferation and the differentiation of a stem cell population. Implementing this approach as a stochastic, single cell-based model allows to follow the progeny of each individual cell and thereby track clonal developments[20]. Depending on the choice of model parameters we can influence how fast the system will converge towards clonal dominance. In particular, if we increase the heterogeneity of the initialized cell clones with respect to their tendency for differentiation (encoded by the standard deviation $\delta$ of the mean differentiation rate) we accelerate the process of clonal dominance (Fig. 3a–c, "equal

clones" vs. "pronounced clonal advantage"). On top of those clonal time series we superimpose several IS per clone. Technically, we sample the number of IS per clone from a Poisson distribution with mean $\lambda$. In GT applications, this parameter $\lambda$ can be easily obtained using the vector copy number (VCN) of the analyzed population, which is an experimentally accessible quantity reflecting the average number of IS per cell. Examples of this superposition for $\lambda = 5$ are shown in Fig. 3d–f.

From these simulated clonal developments, we obtain time series of the relative IS abundance $\hat{I}_i(t)$ which are scalable for their clonal heterogeneity ($\delta$), the average VCN $\lambda$, but also for the number of measurement time points and the level of measurement noise $\sigma$. It is the central advantage of the simulation model, that the ground truth, i.e., the true assignment of IS to clones is intrinsically known and can be used as a benchmark to evaluate the performance of the reconstruction process.

The bioinformatic reconstruction is solely based on the IS time series $\hat{I}_i(t)$, which has been generated from the raw number of sequence reads per IS (see Methods). Briefly, the reconstruction pipeline is initiated by a filtering step to minimize the detection of spurious correlations and to focus on the clones with higher abundance. For the particular example, we are considering the 50 most abundant IS at the final time point of the analysis (Fig. 3g±i). As the second step, we calculated a similarity measure (i.e., the coefficient of determination $R^2$) for all pairs of IS that remained

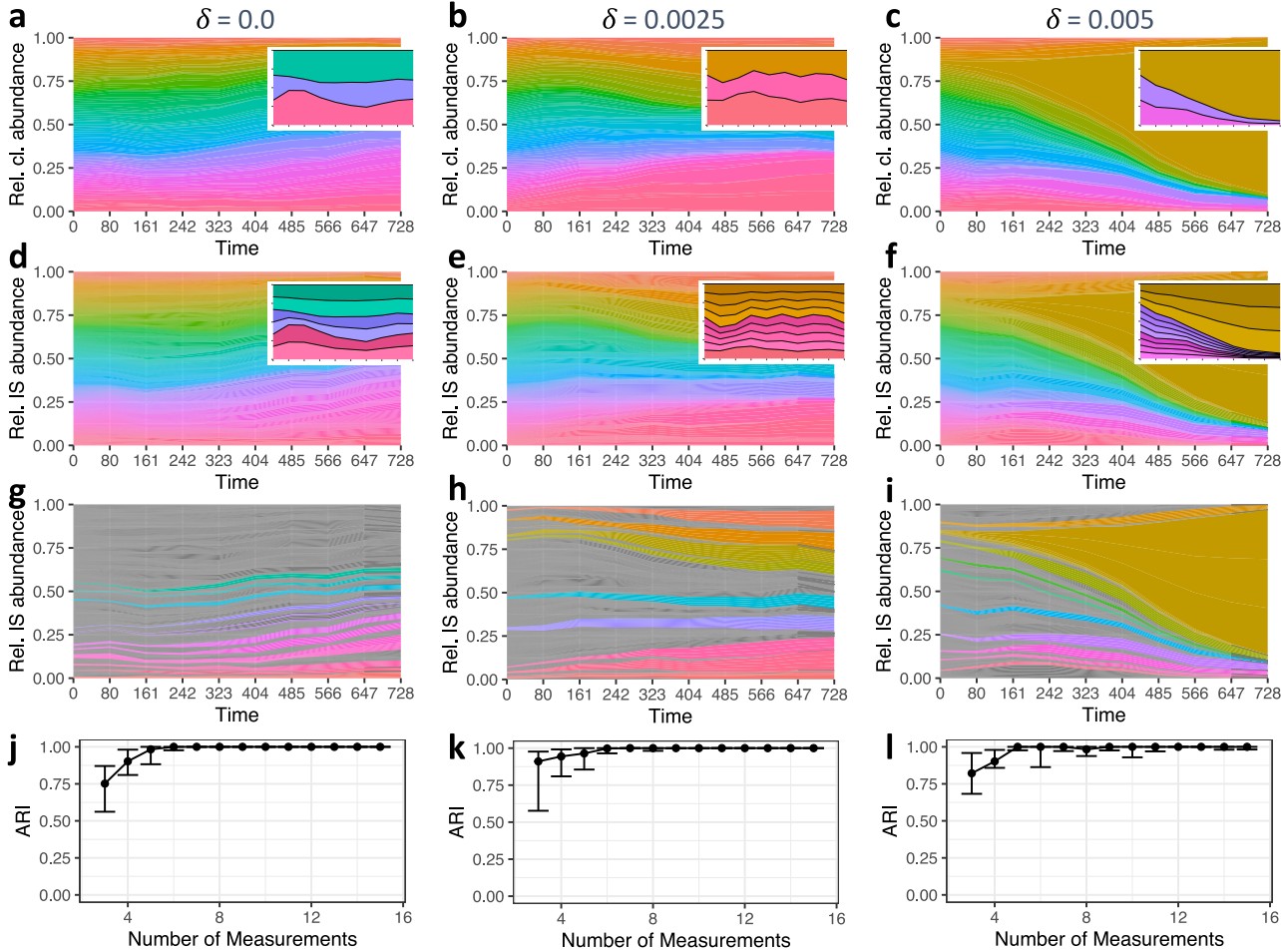

**Fig. 3 Time series of clonal development and their reconstruction based on IS measurements. a**, **b**, **c** Clonal time series (100 initial clones, 100 cells each) for increasing values of the standard deviation $\delta$ of the differentiation rate with an increasing tendency for clonal conversion (left: "equal clones" vs. right: "pronounced clonal advantage"). Insets focus on the three largest clones only. **d**, **e**, **f** Corresponding time series for IS, which are superimposed in each clone. The insets represent the IS substructure of the three largest clones. **g**, **h**, **i** Remaining time series after the filtering step, which limits the analysis to the 50 largest IS being present at the final time point. **j**, **k**, **l** Quality of the clonal reconstruction process as a function of the number of available, equally spaced time points for IS measurements. Reconstruction quality is measured by calculating the adjusted Rand index (ARI) between the known ground truth and the reconstructed clustering (ARI → 1 indicating perfect reconstruction; quantitative analysis is based on 20 independent simulations each; points indicate the median; whiskers correspond to the first and third quartile).

after the filtering step. This mutual similarity is subsequently used to identify clusters of IS with highly correlated behaviors. Those clusters are interpreted as single clones with potentially multiple IS. In a final step, we recalculated the clonal abundances as an average of the IS abundance per clone and for each measurement (Supplementary Note 1, Supplementary Figs. S1 and S2).

We quantify the reconstruction quality using the adjusted Rand index (ARI), which compares the true association of IS to clones with the corresponding assignment obtained from the reconstruction pipeline. Figure 3j, k indicates that the median of the reconstruction quality increases as a function of the number of equally spaced measurement time points, suggesting that a reliable reconstruction can be obtained for most values of the clonal heterogeneity $\delta$ (ranging from no to pronounced clonal advantage) given that at least five independent measurements are available.

Delineating the individual clonal time series according to their assigned clusters indicates that the IS in the same cluster indeed shows highly correlated behavior (Supplementary Figs. S1 and S2). Using this assignment of IS to clones, a reconstructed clonal time series can be obtained which mimics the general behavior

already known as the ground truth. Minor IS that did not pass the filtering step were corrected for the average number of IS per clone and are indicated as background. Supplementary Figs. S3 and S4 further confirm that the reconstruction is unaffected if more but smaller clones are initialized and that even the unlikely case of indistinguishable IS in different clones does not distinctly affect the results.

We further analyzed the influence of the average VCN $\lambda$ and the level of measurement noise $\sigma$ on the reconstruction quality (Fig. 4a–i). We observe that the reconstruction algorithm performs worse for smaller VCN, especially if the average value is as low as $\lambda = 2$ or smaller. In those cases, the majority of clones harbor one IS. As the clustering approach has a tendency to combine weakly correlated IS into one clone, the overall reconstruction quality is diminished. For increased average VCN $\lambda$ a high reconstruction quality is achieved while for very high values ($\lambda > 10$) only few clones remain after the filtering. In an orthogonal dimension, we study the influence of technical noise $\sigma$ on the quantification of each individual IS. Figure 4a–i indicates a robust reconstruction even for increasing levels of measurements noise $\sigma$ (herein, a value of $\sigma = 0.04$ corresponds to

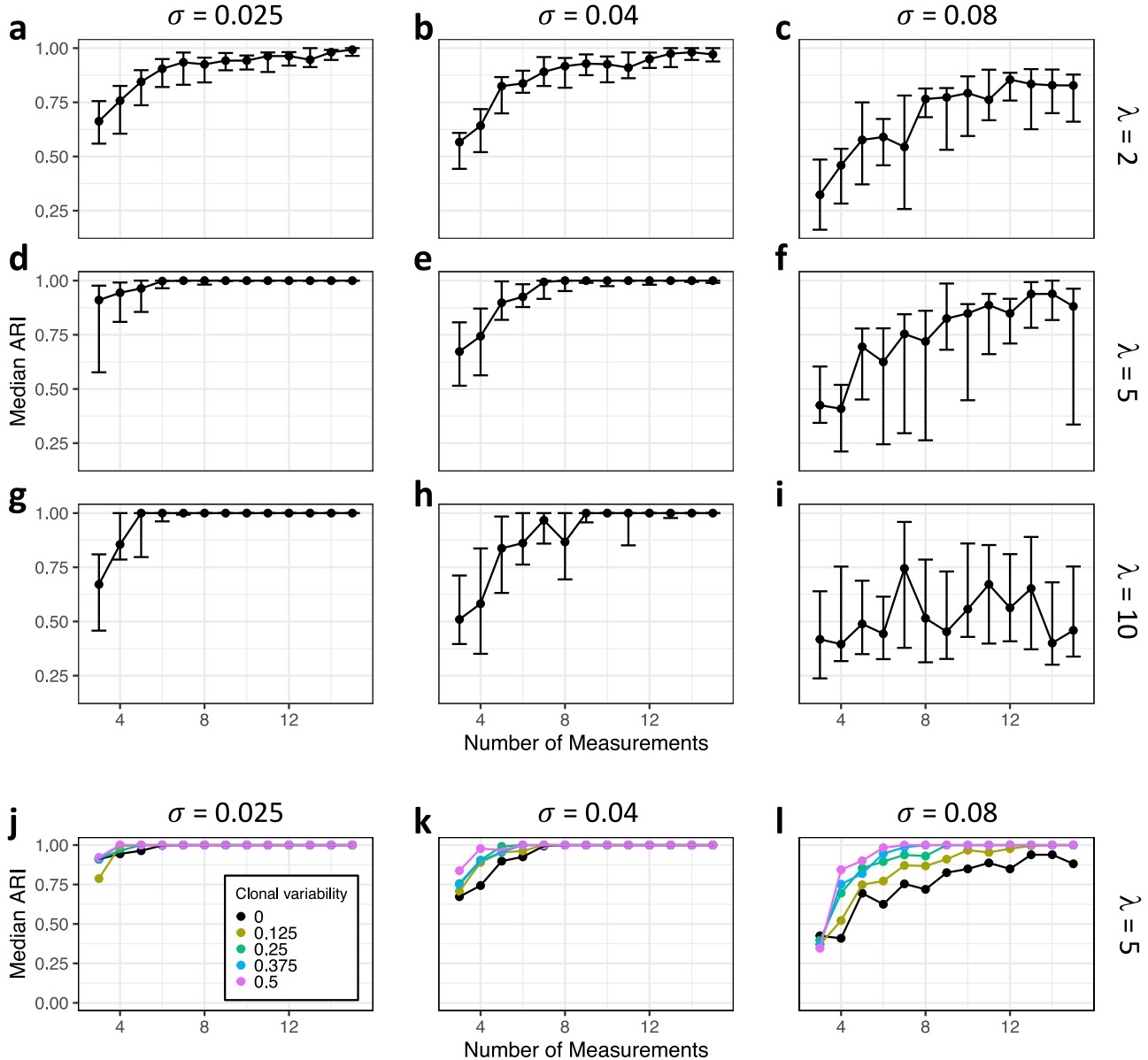

**Fig. 4 Dependencies of the reconstruction quality. a–i** Reconstruction quality with respect to the average number of IS $\lambda$ and levels of measurement noise $\sigma$ for standard deviation of the differentiation rate $\delta = 0.0025$. Rows refer to the average number of IS $\lambda = 2$ (**a**, **b**, **c**), $\lambda = 5$ (**d**, **e**, **f**) and $\lambda = 10$ (**g**, **h**, **i**). The column represents increasing levels of the measurement noise $\sigma = 0.025$ (**a**, **d**, **g**), $\sigma = 0.04$ (**b**, **e**, **h**) and $\sigma = 0.08$ (**c**, **f**, **i**). Each data point is based on 20 independent simulation runs (points indicate the median adjusted Rand index (ARI); whiskers correspond to the first and third quartile). **j**, **k**, **l** Reconstruction quality depends on clonal variability ν. Median ARI is shown as a function of the number of measurements for increasing levels of measurement noise $\sigma = 0.025$ (**j**), $\sigma = 0.04$ (**k**) and $\sigma = 0.08$ (**l**). The coloring corresponds to different levels of the clonal variability ν, which approximates the diversity of available compartments. Each data point is based on 20 independent simulation runs.

a 95% confidence interval of about 8% around the true value). The simulations also indicate that the reconstructions fail for large values of $\sigma$ as true correlations between IS can no longer be detected. A closely similar pattern was obtained when analyzing the number of reconstructed clones compared to the true number of clones that remained after the filtering step (Supplementary Fig. S3). While there is a general tendency to underestimate the true number of clones, this discrepancy diminishes with increasing number of measurements and lower technical noise.

Our results indicate that a correlation-based assessment of IS abundances is indeed suited to identify multiple IS co-occurring in the same clone.

**Assessment of IS abundance in different hematopoietic lineages improves the reconstruction process.** Various studies of the hematopoietic system have shown that HSPC clones obey the tendency to preferentially contribute to one or another hematopoietic lineage[21–23]. For example, some clones preferentially differentiate to T cells while other clones contribute more to granulocytes. For the first case, we would expect to see a prominent contribution of all the clonal IS to the T cells, while the same IS should rarely be seen for the granulocytes. As the analysis of correlations between different IS benefits from varying clone sizes, we hypothesize that the assessment of IS abundance in different hematopoietic lineages can further improve the

reconstruction process. In this notion, measurements of the IS abundance in different lineages are considered as independent samples closely similar to measurements at different time points.

To account for the fact that clones do not equally contribute to different lineages, we introduced an artificial clonal variability in the model simulation, quantified by parameter $\nu$. For $\nu = 0$, the final clone sizes correspond to the clonal time courses (i.e., $I_i^*(t) = I_i(t)$), while for values of $\nu > 0$ clone sizes for each measurement point are randomly and moderately varied, thereby affecting the abundance of all IS for each particular clone.

Clonal reconstruction based on the resulting model simulation for $\nu > 0$ confirms our hypothesis. Figure 4j–l indicates that especially for higher values of the measurement noise $\sigma$ there is a consistent improvement in the reconstruction quality for increasing values of the clonal variability $\nu$. This additional level of variability, which compares to clonal measurements in distinct hematopoietic compartments, appears as a key factor to strengthen the identification of mutually correlated IS and their clonal origin. We explicitly point out, that those measurements do not necessarily require temporal separation, but can be achieved by subfractioning primary samples according to lineage identity prior to sequencing.

We conclude that IS measurements in different hematopoietic sub-compartments can improve the clonal reconstruction process.

**In vitro validation assays confirm the validity of clonal reconstruction**. We validated our reconstruction method using an in vitro experimental assay. To this end we used four K562 cell clones having different and known IS of a self-inactivating lentivirus (SINLV): ID#27 (1 IS), ID#30 (4 IS), ID#37 (6 IS), ID#46 (10 IS) (see Supplementary Note 1, Supplementary Tables S1, S2). The genomic position of each clone-specific IS was previously identified using a well-established *Sonication Linker-mediated* (SLiM)—PCR. In order to replicate a potential in vivo situation and to challenge our clonal reconstruction model, we designed an in vitro assay where the four-cell clones were mixed at different ratios, such that each clone-specific IS was present at a predefined level of abundances (Supplementary Table S1). A second cell line transduced in bulk with a SINLV and having an average VCN = 1.8 was added to all these mixes to generate the background signal of small clones that could be present in a real case scenario and interfere with the detection of emerging clones. True clone abundances within the different mixed populations were confirmed using droplet digital PCR assay and probe-based assays designed for clone-specific IS.

Figure 5a illustrates the measured relative abundance of the prominent IS for the seven different mixes (left bar with black contour). Spurious IS derived from the transduced background (ranging from 7% to 60% for the different mixes) were already removed by filtering for IS with a relative abundance >1% in any sample.

By applying our suggested reconstruction pipeline, we obtained clusters of IS which correlated and are interpreted as *clones* (Fig. 5a, second bar). Figure 5b illustrates the strong inner-cluster similarity (dark green edges), which allows us to correctly identify the IS belonging to three of the four clones. This correct assignment of IS to clones is already achievable if fewer mixes are used for the reconstruction pipeline, indicated by the high level of precision and recall reported for the clones ID#30, ID#37, and ID#46 harboring more than one IS (Fig. 5c). The slightly lower precision for clone ID#46 results from the false assignment of clone ID#27 (which is only characterized by a single IS), also detectable in the network graph in Fig. 5b. It appears as an intrinsic limitation of the clustering approaches that singular IS are preferentially joined to one of the other clusters. Although

heuristic methods (such as the detection of bimodal similarity scores within clusters) can be implemented to detect weakly assigned IS, we recommend a prior visual inspection to identify and compensate for this shortcoming. Correcting for the obvious misclassification in the given case, the in vitro assay confirms that the estimated clonal abundance obtained from the reconstruction pipeline (Fig. 5a, third bar) closely recapitulates the respective ground truth (Fig. 5a, right bar, Supplementary Table S3). The resulting approximation of the true clonal mixture outperforms a sole assessment of the IS which largely overestimates the true number of dominant clones.

Based on these results, we concluded that our method is indeed suited to reconstruct clones from IS using their abundances over different observations.

**Clonal reconstruction for a mouse experiment**. To further confirm the potential of our reconstruction method for the detection of expanding cell clones harboring multiple IS in a polyclonal setting, we took advantage of a preclinical model of HSC gene therapy (GT) based on tumor-prone Cdkn2a$^{-/-}$ Lineage- (Lin-) cells. In this model, eight-week-old lethally irradiated wild-type C57BL6/J female mice ($n = 8$) are transplanted with Lin- cells collected from eight-week old C57BL6/J-Cdkn2a-/- mice ($n = 12$, half male and half female) and transduced at multiple copies by a SINLV expressing GFP (SINLV.PGK.GFP)[24–26]. Due to the tumor-prone background of the Lin- cells, the transplanted animals develop hematopoietic malignancies with a specific kinetic[25] and the transduction with a neutral SINLV allowed the specific marking of the neoplastic expansion and the tracking of its growth by means of IS analyses.

We obtain measurements of IS abundance in different blood lineages (B-lymphocytes (CD19+), T-lymphocytes (CD3+), and myeloid cells (CD11b+)) collected over time at three to five different time points starting from week 4 post-transplantation. Those data are complemented by IS analyses performed on tissue samples obtained at autopsy, where bone marrow, blood, spleen, thymus, and lymph nodes were collected. Following the same reconstruction pipeline as for the simulation scenario, we started off by filtering the most abundant clones having a relative abundance of more than 1% in the final blood or tissue samples (Fig. 6a). Analyzing the mutual similarities between those dominant IS, we identified those with a closely correlated behavior, which appear as strongly connected clusters in corresponding network maps (Fig. 6b). We interpret this pronounced, visual separation of clusters as a strong indicator of common clonal origin. For the particular example, our analysis suggests that 53 IS that remained after filtering can be optimally mapped to eight different clones. Representations of the clustered time series confirm the high inner-cluster similarity (Fig. 6c).

Based on this assignment of IS to a much smaller number of clones we can now translate the IS time series into a corresponding clonal time series. Figure 6d indicates a less polyclonal pattern as suggested by just accounting for the IS. Figure 6e further acknowledges less abundant IS that did not pass the initial filtering step (indicated in gray), leading to a quantitative description of dominating *clonal* dynamics over the full experimental period. Further examples from the same experimental setting are provided in Supplementary Figs. S6–S9.

Our analysis of an experimental, polyclonal GT setting with known integration of multiple vectors per cell clone illustrates the applicability of our method in an application-relevant context.

**Clonal reconstruction for a nonhuman primate data set**. Macaques are important animal models to study safety aspects of GT and to assess long-term effects of transduced HSPCs[8,27,28]. Recently, a report was published documenting the rare case of

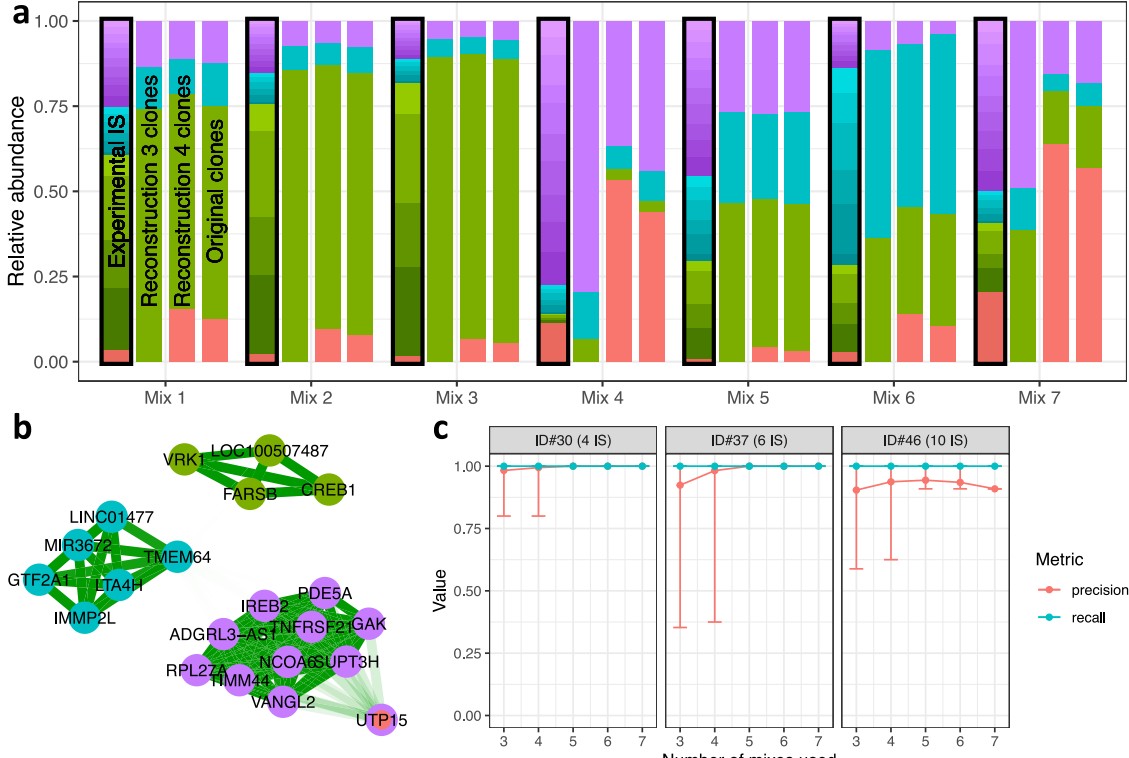

**Fig. 5 Analysis of in vitro mixes of the validation assay. a** For each of the seven analyzed samples (mixes 1 to 7) four stacked bar plots display from left to right: the relative proportion of measured IS after filtering (black contour, highlighting the available data in an experimental/clinical context, although the assignment to one of the four clones (i.e., the coloring) is generally unknown), the relative proportion of the different clones after applying the reconstruction routine, the relative proportion of the different clones after manual correction (see text), and the relative proportion of the four original clones. **b** Network map indicating the clustering of mutually similar IS. Shading of the edges indicates the mutual similarity $R^2$. The inner color of each node represents the true assignment to one of the four clones, while the outer ring corresponds to the result of the reconstruction process. **c** Precision (true positives/(true positives + false positives)) and recall (true positives/(true positives + false negatives)) for the three clones ID#30, ID#37 and ID#46 containing 4, 6 and 10 IS, respectively. All available permutations for the different number of mixes (ranging from 3 mixes to the maximum number of 7 mixes, filtering for IS with a relative abundance > 0.5% in any of the selected mixes) were used to evaluate the benchmark parameters for the individual clones. Whiskers indicate the minimum and maximum values observed.

abnormal, fatal clonal hematopoiesis detected in one of these animals previously transplanted with lentivirally transduced stem cells[29]. We assessed the respective data set reporting IS abundances in four different hematopoietic compartments (T-lymphocytes, B-lymphocytes, granulocytes, monocytes) and at nine different time points post transplantation.

Filtering for IS with a relative abundance of more than 1% in any of the final samples we identify 14 larger IS which our suggested algorithm clusters in three distinct clones (Fig. 7a–c). Clearly, there is one clone (Fig. 7d, shown in red) that progressively dominates the time course and finally accounts for almost 100% of the granulocytes and monocytes (Fig. 7e). Our findings confirm the primary results[29], in which the same six continuously detectable integrations could be assigned to this dominating clone. Furthermore, we also observe that the other two clones appeared with a fluctuating abundance over time, and primarily contributed to T-lymphocytes. The observation that the three clones contribute differently to the different hematopoietic sub-compartments further supports our earlier findings that IS measurements in different compartments can improve the clonal reconstruction process. Technically, it is equally interesting that the (almost) correct association of the six "red" IS to the same clone could have already been drawn much earlier. Supplementary Figs. S10, S11 indicate that already at day 187 and day 266 this association would have been feasible. This analysis illustrates the potential of our suggested method for the early detection of expanding clones.

**R implementation: the *MultIS* package at CRAN**. In order to facilitate the application and further development of our analysis pipeline we provide a corresponding R package named *MultIS* via CRAN (https://cran.r-project.org/package=MultIS). Starting out from a list of IS abundances $I(t)$ for multiple measurements, the package provides all individual functions for the analysis pipeline along with graphical representations (Supplementary Note 4). We provide a corresponding R-script file to reproduce our analysis and figures for this manuscript in a separate repository (https://gitlab.com/imb-dev/clonal-reconstruction-figures).

**Discussion**

The unbiased assessment of temporal clonal contributions to different hematopoietic compartments is limited by the co-occurrence of several IS within the same clone which is difficult to resolve experimentally. In this work, we propose a bioinformatic pipeline that overcomes this limitation and leverages intrinsic correlations between IS abundances derived from the same clone to recover the true clonal structure. Our approach relies on the idea that IS from the same clone appear in the same relative frequency across time-series measurements and among different hematopoietic cell types, while this correlation is missing for IS resulting from different clones. This concept is translated into a corresponding analysis pipeline, which we provide as a publicly available R package *MultIS*.

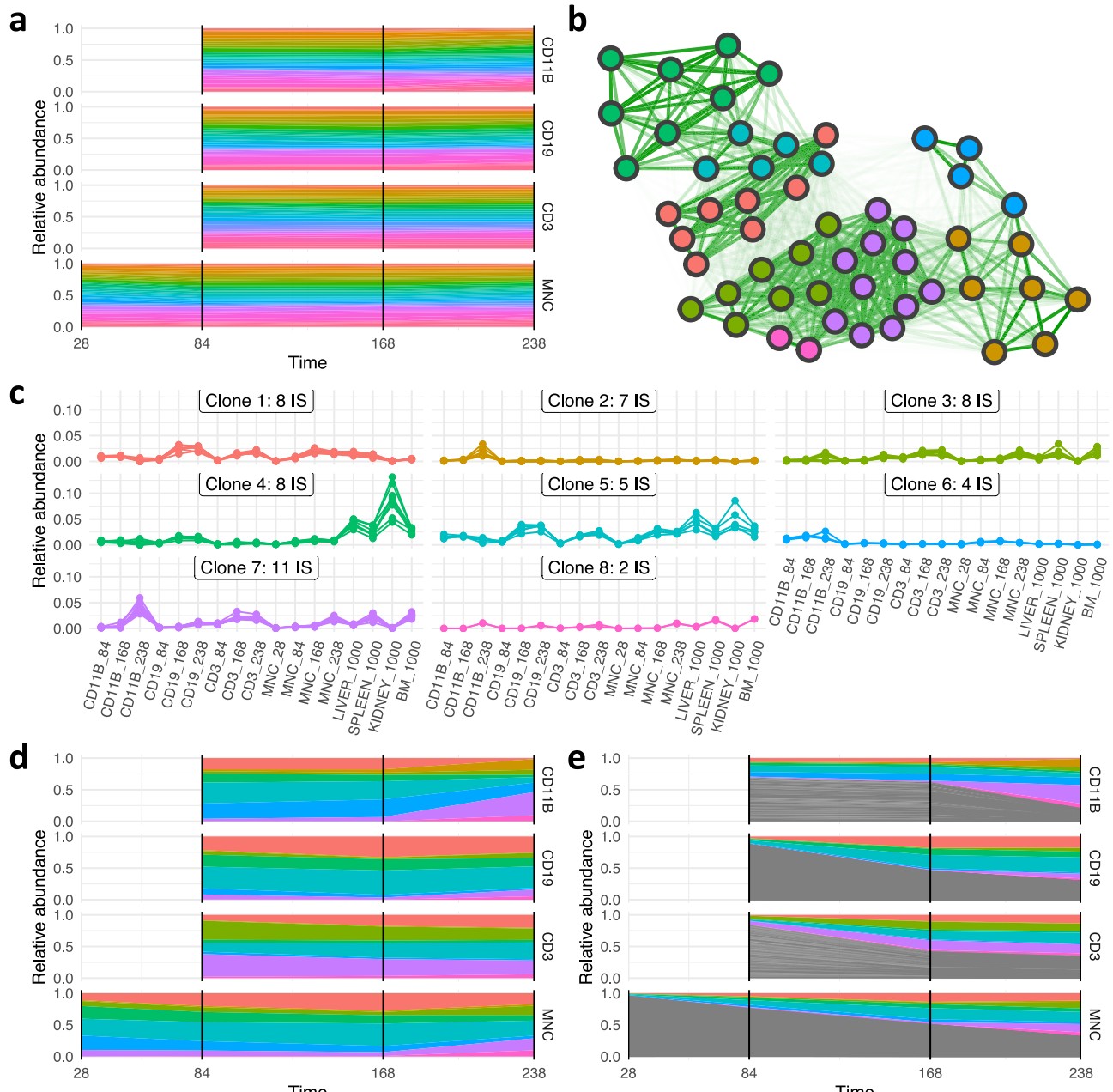

**Fig. 6 Experimental data and clonal reconstruction for mouse pool E4C. a** Relative abundances of IS as a function of time for subsets of CD11b, CD19, CD3, and mononuclear cells, for which multiple measurements are available. **b** The network map depicts the similarity between each pair of IS (indicated by edge brightness) superimposed by the optimal clustering obtained from the reconstruction pipeline (indicated by color of the nodes). **c** Time series of all IS assigned to the same clusters for all available compartments and time points (preserving the color scheme). **d** Corrected clonal time series for the eight identified clones. **e** Corrected clonal time series together with the IS that did not pass the initial filtering step (indicated in gray).

We first optimized and validated our method using mathematical simulations of clonal dynamics. We demonstrated that the method is broadly applicable for settings with VCN $\geq 2$ in the dominating clones. Systematic variation of central model parameters identified fewer measurements and increasing measurement noise as limiting factors of the reconstruction process, as pairwise correlations of IS abundance are harder to detect. In practical terms, we observed reliable clonal reconstruction using eight or more measurements. While an increasing measurement error limits the reconstruction quality, this effect can be compensated by considering different hematopoietic sub-compartments with varying clonal contributions, even if they are measured at the same time point. Our simulation studies confirmed that the central source of information for the reconstruction process is based on the variability in IS abundances re-captured at different measurements. Intuitively, if all clones are always present with a highly reproducible proportion, also the IS of these clones is more or less constant over repeated measures. However, a reliable correlation can only be obtained if IS of the same clone vary in a synchronized manner, namely due to their changing clonal abundance. This observation indicates that not only temporally separated measurements but also IS analyses in different hematopoietic lineages with distinct clonal contributions at the same time point are very valuable to reconstruct the IS affiliation with good precision.

Applying the reconstruction pipeline to a set of in vitro assays in which cell clones with different numbers of IS are mixed in

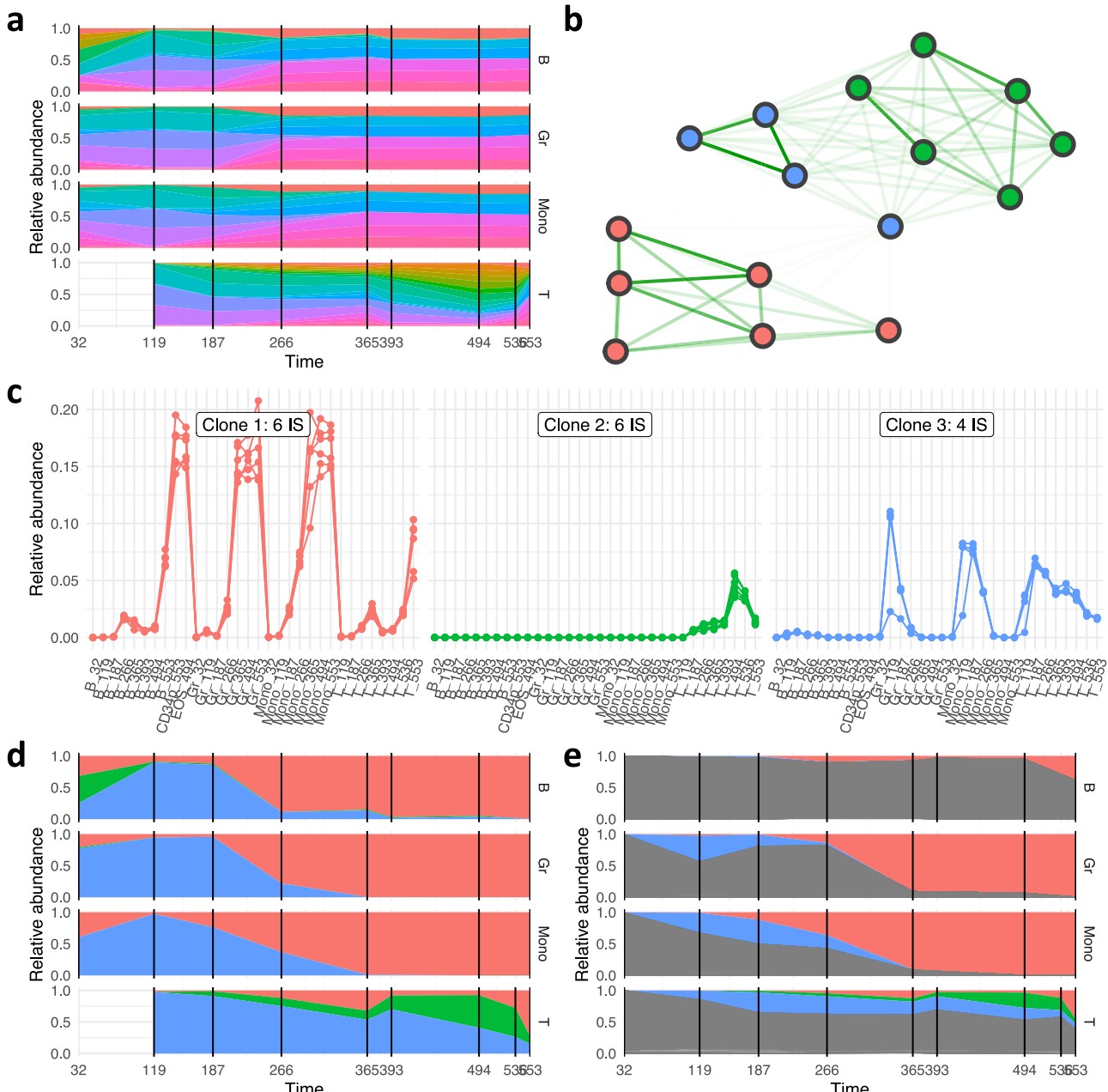

**Fig. 7 Experimental data and clonal reconstruction for a rhesus macaque.** The animal received an autologous transplant in which the HSPCs were transduced with a lentiviral vector containing a strong constitutive promoter-enhancer. The respective data for IS abundance in different compartments and at different time points is available from ref. [29]. **a** Relative abundances of IS as a function of time for subsets of cells (T-lymphocytes, B-lymphocytes, granulocytes, monocytes), for which multiple measurements are available. **b** After filtering for all IS with a relative abundance >1% at the final time point (553 d), 16 IS were further analyzed. The network map depicts the similarity between each pair of IS (indicated by edge brightness) superimposed by the optimal clustering obtained from the reconstruction pipeline (indicated by color of the nodes). **c** Time series of all IS assigned to the same clusters (preserving the color scheme). **d** Corrected clonal time series for the three identified clones. **e** Corrected clonal time series together with the IS that did not pass the initial filtering step (indicated in gray).

predefined ratios confirms the feasibility of the approach. Solely based on the quantification of IS abundance we can identify which of the IS belongs to which clone. These experiments also point towards a limitation of the automated clustering approach, as it has a tendency to misinterpret clones with single IS. The clustering has a higher tendency to join such IS to other clusters instead of labeling them as unique. Wrong assignment of a few unique IS to identified clusters will only moderately affect the overall interpretation of the clonal reconstruction, although we strongly recommend a visual inspection of the clustering results

to identify obvious misclassifications. Threshold-based corrections (using e.g., the detection of a bimodal distribution of similarity scores within one cluster) can heuristically target this problem, while critical cases might need to be resolved by an experimental validation based on the sequencing of single-cell colonies.

We further used the reconstruction pipeline to estimate clonal expansion in an in vivo model for malignant hematopoiesis. Our results confirm the hypothesis from the simulation results, namely that the assessment of different hematopoietic lineages

improves the clonal reconstruction process. Although the true assignment of IS to clones is unknown for these experiments, the overall similarity and consistency of the clustered time series reinforce our primary intention that the reconstruction process can also be applied to in vivo settings with rather stable polyclonal configurations. These approaches have been complemented by applying the reconstruction pipeline to time-course data from a non-human primate model in which the expansion of a clone with multiple IS was observed[29]. Our suggested method successfully revealed the correct association between the IS of the dominating clone, thereby demonstrating the reliability and precision of our approach. Moreover, our results showed that this association is already feasible at a much earlier time point in a continuous monitoring process, highlighting the potential of our method for applications in GT.

We acknowledge that the suggested and necessary filtering step may lead to incomplete assignments of IS to clones. However, rather than identifying all IS within one clone, we focus on the identification of how much-unaccounted co-occurrence of IS in the same clone biases the overall assessment of clonality. Trying to detect early signs of a developing clonal dominance, it is plausible to control how much the largest detected IS are truly independent clones or whether correlations in their abundance may indicate a common clonal origin. Inclusion of additional, minor IS that were neglected in the filtering step does not substantially change these conclusions, even if the IS would add to the already identified clones. The same applies for the unlikely case that the same IS occurs independently in two clones as the larger contribution would prevail and be correctly correlated to the dominating clone.

Current studies of clonal dynamics only quantify IS abundances relative to the total observed IS and interpret those time series as independent clones. We showed that this approach can clearly lead to an overestimation of the number of clones and an underestimation of their relative contribution. However, the quantitative assessment of clonal behaviors is crucial for the evaluation of the safety of GT as well as for the interpretation of experimental studies to understand physiological hematopoiesis and related malignancies or to study hematopoietic reconstitution in vivo. The availability of repeated IS measurements at different time points as well as in different hematopoietic cell types, especially in a clinical context, represents the optimal data basis necessary for the successful application of our suggested methodology and to reach a more thorough understanding of temporal clonal developments.

## Methods

**Ethical approval**. All experimental procedures were performed according to protocols approved by the Animal Care and Use Committee of the San Raffaele Institute (IACUC 859) and communicated to the Ministry of Health and local authorities according to Italian law.

**Reconstruction pipeline**. The reconstruction pipeline provides a method to identify clones based on the tracking of IS over time and/or in different hematopoietic compartments. An initial *filtering* step (Fig. 2a) restricts the analysis to prevent potential biases from under-represented clones. For the filtering of the simulated time series, we obtain interpretable results over a broad range of parameter settings if we consistently consider the 50 largest remaining IS at the final time point. For the biological data, we filter IS with an abundance above 1% at the final measurements to account for conservative detection thresholds of the quantification methods. Next, we use a regression approach to calculate the coefficient of determination $R^2$ quantifying the pairwise *similarities* among all IS (Fig. 2b), which are represented as a similarity matrix $S$. The resulting similarities feed into the subsequent *clustering* algorithm (partition around medoids ("PAM")) that identifies co-occurrent IS by their abundance under the hypothesis that different IS generated by the same cell clones share the same quantification (Fig. 2c). Since the real number of clones to be detected is unknown, we apply the similarity clustering algorithm for all sensible number of clusters k and select the best solution by comparing the cluster qualities (using a silhouette score[30] accounting for high

inner-cluster similarity and low outer-cluster similarity). In a final step, *clonal abundances are reconstructed* by accounting for the average number of IS identified per clone (Fig. 2d). To quantify the similarity of two clusterings, especially when comparing a known ground truth with the results of the reconstruction pipeline, we use the ARI[31,32].

In the context of the suggested filtering step, there are no relevant computational constraints for the suggested pipeline (computation time in the range of seconds, compare Supplementary Fig. S12). Further technical details are provided in Supplementary Note 1.

**Simulation data**. In order to test and validate the clonal reconstruction pipeline within a scalable context, we generated in silico data based on a corresponding and published mathematical modeling approach to describe clonal tracking data[20]. It is the advantage of this simulation model that the true association of IS to clones ("ground truth") is intrinsically known. Technically, the time courses are generated from a stochastic, single cell-based model of clonal dynamics in which cells of different clones proliferate and differentiate at rates that are kept fixed for each clone. The differentiation rate for each clone c is initialized from a normal distribution $\mathcal{N}(d, \delta^2)$, while the proliferation rate is dynamically regulated by a logistic growth function with overall carrying capacity $K$ and maximal proliferation rate $p_{max}$ (which is identical for all clones, see Supplementary Table S4 for parameter values). Clone sizes $N_c^{raw}(t)$ are given by the absolute number of cells belonging to a clone c at time point t.

On top of the clonal dynamics, we initially assign a certain number of unique IS for each clone c according to a Poisson distribution $Pois(\lambda)$ in which λ reflects the average number of IS for the particular transduction setting. The raw abundance $I^{raw}$ (i.e., without any measurement noise) of each individual IS i belonging to a clone c at time t is given as

$$I_i^{raw}(t) = N_c^{raw}(t) \tag{1}$$

In order to account for the *measurement noise* of the detection process, we add a multiplicative noise $g_i(t) \sim \mathcal{N}(1, \sigma^2)$ to the readout of every IS measurement

$$I_i(t) = g_i(t) \cdot I_i^{raw}(t) \tag{2}$$

to obtain the time courses I(t) for all IS.

In order to mimic changing clonal contributions to different hematopoietic cell types, we introduce an additional inter-clonal shift, termed *clonal variability* ν that superimposes a random factor $f_c(t) \sim \mathcal{N}(1, v^2)$ at each time point of measurement t, such that the abundance of simulated IS $I^*(t)$ is given as

$$I_i^*(t) = f_c(t) \cdot g_i(t) \cdot N_c^{raw}(t). \tag{3}$$

Herein, the first factor is clone specific and affects all IS of the same clone, while the second factor accounts for an individual measurement error for each IS. Following the above motivation, the temporal dimension t in the simulated data reflects both time *and* cell type in an in vivo study. The *relative* IS abundance is calculated as

$$I_i^{*rel}(t) := \frac{I_i^*(t)}{\sum_l I_l^*(t)} \tag{4}$$

We varied the average VCN λ, the clonal variability ν, the measurement noise σ, as well as the number of measurements to generate a range of different model realizations for which the reconstruction process has been tested. Further technical details are provided in Supplementary Note 2.

## Experimental data

*SLiM-PCR for IS quantification*. Genomic DNA containing respective vector integrations was sheared by sonication, end-repaired and adenylated, then ligated to a barcoded linker cassette and subjected to two different rounds of PCRs allowing the amplification of the cellular genome close to the vector IS. PCR amplicons are then assembled, sequenced, and processed by dedicated bioinformatic pipelines VISPA2[33] to identify the different IS for each sample[7] (see Supplementary Table S2, Supplementary Notes 3 and 6). The sensitivity and precision of the SLiM-PCR approach have recently been described in an in vitro limiting dilution assay, benchmarking the limit of detection at 0.1% for a single IS and measuring the signal-to-noise ratio[7].

*Validation assay*. The validation assay uses cell clones from the K562 cell line (derived from a chronic myelogenous leukemia patient in blast crisis) and the Jy cell line as background (an Epstein–Barr virus immortalized B cell lymphoblastoid cell line) that were available in the lab. We prepared a set of seven samples containing variable proportions of four different cell clones (see Supplementary Table S1) mixed into the background of transduced cells. Each of the four clones is characterized by a different number of IS (ID#27 (1 IS), ID#30 (4 IS), ID#37 (6 IS), ID#46 (10 IS)), while for the background only the average number of IS was determined to be ~1.8 copies. The prepared mixtures were primarily intended to address issues of sensitivity and reproducibility, while the variable abundance of four different clones serve as a suitable sample to verify whether the suggested bioinformatical pipeline correctly assigns the IS to their respective clones. For all samples, LV/genomic junctions were retrieved by SLiM-PCR, sequenced using

NGS technologies, and mapped on the human genome to identify the nearest RefSeq gene. The relative amount of each clone in the mix was quantified and confirmed by droplet digital PCR (ddPCR) assay where primers and probes were specifically designed for at least one of the LV/genome junctions of each clone. As a filtering step, we are only considering IS that present with a relative abundance of >1% in any sample, thereby excluding IS from the background of other transduced cells.

*Clonal tracing in mice*. We used data from a set of mouse experiments in which wild-type recipient mice were transplanted with *Cdkn2a*$^{-/-}$ BM-derived lineage-negative cells (lin-, HSPCs equivalent) carrying multiple vector integrations (average VCN $\lambda$> 10, Supplementary Table S5). The transduction and transplantation strategy used for these experiments replicates the procedure reported earlier[24,25]. Briefly, Lin- cells were collected from eight-week-old C57BL6/J-Cdkn2a-/- mice (*n* = 12) and transduced at a Multiplicity of Infection 100 with the previously described self-inactivating lentiviral vector (SINLV) expressing GFP (SINLV.PGK.GFP). Transduction efficiency was evaluated by FACS analyses at 6 days post infection and reached 90% GFP-expressing cells. The day after transduction, eight-week-old lethally irradiated wild-type C57BL6/J mice (*n* = 8) were transplanted with vector-transduced cells (5–7.5 × 10$^5$ cells/mouse) by intravenous tail vein injection. Blood samples were taken at three to five different time points post transplantation, pooled from two to three animals and FACS sorted according to the phenotype of B- (CD19+), T- (CD3+), mononuclear, and myeloid (CD11b+) cells (Supplementary Fig. S13). Furthermore, at autopsy, the bone marrow, blood, spleen, thymus, and lymph nodes from each mouse were collected for IS site retrieval, providing the IS of the dominant/tumoral clone infiltrating the different tissues. In all these samples, LV/genomic junctions were retrieved from DNA samples by SLiM PCR, sequenced, and mapped on the mouse genome to identify the nearest RefSeq gene. VCN was measured by ddPCR for most assays and correlate well with the number of IS obtained from *MultIS* (Supplementary Table S5). Further experimental details are provided in the Supplementary Note 3.

**Statistics and reproducibility**. Statistical analysis was limited to deriving point estimates (mean, median) and corresponding variances using R version 4.0.5. The coefficient of determination $R^2$ between pairs of IS was calculated using the function "summary.lm" from the same R version. No sample size estimates were required for the development of the analysis method.

**Reporting summary**. Further information on research design is available in the Nature Research Reporting Summary linked to this article.

## Data availability

All datasets analyzed within the current study are available from the clonal-reconstruction-figures repository at https://gitlab.com/imb-dev/clonal-reconstruction-figures/-/tree/master/data. The macaque data set[29] can be obtained from the GEO database, accession number GSE153130. Further details are provided in Supplementary Note 5.

## Code availability

The code to reproduce all figures and analyses in this manuscript is available from the clonal-reconstruction-figures repository, https://gitlab.com/imb-dev/clonal-reconstruction-figures. The corresponding scripts in the repository use our R package *MultIS*, which is available via CRAN, https://cran.r-project.org/package=MultIS.

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

## Acknowledgements

We thank Thomas Zerjatke, Lars Thielecke, and Friedemann Uschner for critical discussions of our manuscript and Monica Volpin for sharing reagents. This work was

supported by the German Federal Ministry of Research and Education (BMBF, grant number 031A315 "MessAge") to I.G. and Telethon Foundation TGT16B01 and TGT16B03 to E.M. and Giovani Ricercatori Grant 2016 from the Italian Ministry of Health to A.C. and D.C. (GR-2016– 02363681).

## Author contributions

S.W., C.B., and I.G. conceived and designed the study. S.W., C.B. developed computational tools and performed analysis. L.R. and P.G. performed the experiments under the supervision of D.C. and E.M. A.C. and D.C. analyzed the experimental data. S.W., C.B., D.C., A.C., and I.G. wrote the paper. All authors reviewed the final manuscript.

## Funding

## Competing interests

The authors declare no competing interests.
