## [Peer Review File · Nature Communications]

Reviewers' Comments:

Reviewer #1:

Remarks to the Author:

The revised paper is significantly improved from the previous version that I reviewed for another Nature journal. I have only one remaining suggestion. It seems the current clustering algorithm assumes the same integration site occurs at a single time point. When many integration sites are observed, is it possible that the IS is integrated into indistinguishable, proximal locations in different clones? How this will impact the clustering results based on the abundance patterns of IS insertions? It will be useful to study this impact in the simulation experiments.

Reviewer #2:

Remarks to the Author:

In my previous comments, I mainly raised the concerns about methods and descriptions. In the revised manuscript, the authors answered these concerns by updating the manuscript and supplementary methods. Overall, it is greatly improved. I have no further concern.

Reviewer #3:

Remarks to the Author:

I would like to thank the Authors for working on revising their original submission (reviewed in Nature Computational Science), including the revisions made in the repository. I now get a better understanding of the computational challenges of the problem that MultIS aims to solve.

I emphasize that my primary expertise is in Computer Science and Math so my review is focused on these aspects of this work. My main focus is on the following:

- (i) assessing novelty of the proposed approach, in particular with respect to the previously published clustering algorithms.
- (ii) assessing its correctness from the mathematical point of view and its contributions from the computational and mathematical points of view.

After having close look into the methods presentation in the main text, as well as in the supplementary, I have a single major comment/question: what is the main algorithmic novelty of this work? Based on what I could see, it seems to be very minimal in order to justify a publication in a journal Nature Communications. Please note that I am not claiming at all that the package itself is useless and I actually leave a judgement about that to the Reviewers with stronger biological background and more experience in working with this type of data.

I am also slightly concerned that the Authors are having some misunderstanding of certain tools designed for analyzing cancer sequencing data. For example, in the Introduction (Page 3) they mention that PyClone aims to recover the underlying clonal phylogeny, which is not correct (see the cited paper from Nature Methods: Roth et al. 2014). Namely, PyClone does not recover any phylogeny. It just performs clustering based on the read counts (corrected for purity and ploidy). I recommend a careful revision of these statements. Some possible alternatives to "clonal phylogenies" can be "clonal composition" or "clonal structure".

In their response to point 3 raised by Reviewer 1, the Authors claim that "the probability of observing two independent events in the same position follows a binomial distribution..." and

based on that assumption they obtain extremely low probability for having such events. Similar (but not fully equivalent) conclusions have been repeatedly made and then questioned in the studies of cancer. Therefore I would be a bit more careful with making so explicit statements about which particular distribution we are having in this case. I would rather focus on showing that a couple violations of the assumption do not have a significant impact on the results of the method. I understand that this might be hard to do on the real data, but it should be doable in the simulated setting. Related to this, I also recommend having a short look at the following recent paper <https://www.nature.com/articles/s41588-021-01005-8>.

MINOR:

Now that I better understood the key difference between analyzing tumor and IS data, looking only into Figure 1, I would still be convinced of the applicability of methods for analyzing multi-timepoints tumor sequencing data. However, it might indeed be challenging to come up with a better visualization so I do not necessarily expect any additional updates to this figure, especially considering that there is now some discussion of the tumor data clustering algorithms in the main text.

Reviewer #4:

Remarks to the Author:

Review: "Clonal reconstruction from co-occurrence of vector integration sites allows accurate quantification of expanding clones in vivo"

Sebastian Wagner, Christoph Baldow, Andrea Calabri, Laura Rudilosso, Pierangela Gallina, Eugenio Montini, Daniela Cesana, Ingmar Glauche

Here Wagner et al. detail a method to quantify the effects of vector copy number when performing viral integration site (IS) analysis in the hematopoietic stem and progenitor cell gene therapy (GT) setting. Multiple studies have demonstrated that two general characteristics of virally-transduced cells can result in increased risk of insertional mutagenesis and subsequent malignant transformation: First, the locus of integration, such as an integration event which activates a proto-oncogene or silences a tumor suppressor gene (i.e. integration in or near a "high risk" genomic locus). Second, the number of individual integration events which occur in a single cell. This is based on the demonstration that multiple integration events may occur in single cells during viral transduction. The higher the number of genomic perturbations, the higher the risk of altered genomic stability. While much of the field has focused on identification of IS associated with high-risk genomic loci, the ability to identify cells with multiple IS events has received far less attention. Thus, there is a clear need to develop better methods for estimating clonal dominance even when masked by high vector copy numbers per cell.

In current analysis pipelines, the genomic site of (semi-random) provirus integration is used as a cellular barcode to track clones after transplant. However, this approach treats each identified IS as a unique clone because the present sequencing methodology does not include cellular identifiers which can attribute IS to the cell from which they came. In the case where multiple integration events occur in one cell, these IS will be inaccurately called unique clones in followup procedures. This could lead to an underestimation of clonal dominance and, importantly, could result in delayed identification of clones with transformative potential.

A process or pipeline to identify multiple IS belonging to a single clone would be highly useful for advancing our understanding of clonal dominance and potential for untoward cellular transformation. Here the authors focus on a pipeline analysis which could be applied to existing IS data for identification of unique IS with high probability of being present in the same cell. This analysis is dependent on the method of IS analysis capturing an accurate representation of clonal contribution over time and, the largest clonal pools differing enough in size and/or differential contribution to blood lineages in hematopoiesis. Under these conditions, a clustering algorithm could be used to group IS that are detected similar in quantity and lineage contribution over time into units that represent individual clones. While this methodology is timely and has great potential, the presented experiments could be greatly strengthened by the items mentioned below.

We would consider the manuscript of high value to the field and readership of Nature Communications if these concerns were addressed in a revision.

Major Revisions Requested:

1. Detection of aberrant hematopoiesis in GT setting at the earliest possible time is clearly important. The ability to predict aberrant hematopoiesis before clinical emergence is the holy grail in GT, but to do this requires careful study of existing cases of insertional mutagenesis as a result of multiple IS per cell. A data set of clone tracking in a GT setting in hematopoietic stem and progenitor cells where a single clone with multiple IS became dominant exists¹. It would strengthen this paper if this algorithm could be run on this existing dataset and determine the earliest point where the multiple IS from the dominant clone could be identified as a single clone. Alignment of this detection with the clinical data from this subject would help determine how much earlier risk could have been identified. This would serve as a real-life proof-of-concept for the utility of the presented algorithm.

2. The above algorithm check in the setting of a documented, multiple IS clone which displayed insertional mutagenesis is one proof-of-concept. Key to demonstrating a useful analysis pipeline is estimating how often the analysis incorrectly identifies multiple-IS clones (false positive call rate) and misses relevant multiple-IS clones (false negative call rate). The authors could demonstrate in an experimental setting with PCR confirmation of individual clones, the frequency of correct multiple IS calls in a given dataset. This could be done with hematopoietic colony-forming cells in an in vitro setting, which takes just a few weeks to generate, and could be compared to VCN from the same dataset of individual colonies.

3. The need to understand the effects of multiple IS in individual clones is clearly valid and clustering IS based on similarities of quantity and lineage contribution seems correct. I think the validation on mouse and fixed populations were good. In the mouse experiment were any of the VCN numbers from simulation verified? As noted in point 2, connection of VCN and algorithm-called multiple-IS clones seems like an important check to perform, at least in an in vitro setting. Another option would be to use publicly accessible data from a study where VCN on single colonies was performed and run the algorithm on this dataset.

4. The assumptions made about a clone population in a non-human primate (NHP) or human setting are concerning. The algorithm for predicting HSC clonal dynamics predicts the generation of pools of HSC clones that happen to replicate and not differentiate. For a low number of clones, (e.g. Figure 3), using parameters of this model that assume inheritable differences in clone growth rate and clone lineage bias makes clones easily distinguishable. However, both assumptions may be optimistic for the case of clone identification. For the case of an NHP or human transplant, the initial number of clones is likely to be 100,000-1,000,000 or more. Published hematopoietic engraftment and contribution dynamics in NHP and humans suggest this HSC diversity will drop rapidly to 1,000-10,000 unique clones, each represented by a large pool of HSCs. Clustering will in part depend on differences in clone pool size, but many of the top pools will have very similar size, especially if no parameter is added to give some clones a growth advantage.

The idea that clones have a built-in bias of the lineages they produce is also optimistic. To date, IS processing has not been done to a level that proves clonal bias. If these assumptions are taken out of the "ground truth" model and a more realistic clone population is simulated, would the clustering still be accurate?

Two simulations with identical starting condition that are close to an actual GT application would be helpful. In one case no clone has a growth advantage. Using this case, are IS incorrectly clustered

together? In the next case if one high VCN clone is given a growth advantage, are these IS identified as one clone and at what time point does this become apparent?

5. This process is dependent on the reliable capture of IS over time. In the introduction section, a value of 0.1% is given for the reliable quantification of IS abundance. However, the threshold for

reliable quantification of IS abundance will depend on gene marking level of the test organism, and the methodology applied to identify IS, which can vary from group to group. At minimum, we would like to see a review of the available study data and an estimated range of this value based on reported data, with simulations that cover at least the two extremes and mean of this range.

a. For example, in an autologous transplanted NHP, gene marking levels in peripheral blood white blood cells may be 30% or higher and if a fluorescent reporter gene is included in the viral vector, flow sorting could be used to purify IS containing cells. This means in a typical sample (500,000 genomes), each IS at 0.1% would be present at 500 copies. IS capture efficiency has been demonstrated to be low (~5%).² This would result in an average of about 25 good IS sequence reads. If the gene marking level is low, such as ~5% as has been demonstrated in the human GT setting, then a 0.1% clone would be present in $500,000 * 0.05 * 0.001 = 25$ copies. If IS capture is about 5% then this would mean an average of 1.25 sequence reads for this clone. This is concerning for reliable detection. For groups who want to apply this algorithm, what are the parameters of gene marking and IS quantification accuracy that will generate a reliable result? For clones detected, whatever their VCN state, at what level of contribution do they become concerning?

References

1. Espinoza, D.E, Fan,X...Dunbar,C.E. (2019) Aberrant Clonal Hematopoiesis following Lentiviral Vector Transduction of HSPCs in a Rhesus Macaque, *Molecular Therapy* , Volume 27, Issue 6,5 June 2019, Pages 1074-1086.

2. Adair J.E., Enstrom M.R., Haworth K.G., Schefter L.E., Shahbazi R., Humphrys D.R., Porter S., Tam K., Porteus M.H., Kiem H.P. (2021) DNA Barcoding in Nonhuman Primates Reveals Important Limitations in Retrovirus Integration Site Analysis. *Mol. Ther. Methods Clin. Dev.* 2020;17:796–809.

Point-by-point response to the reviewers' comments

We thank the reviewers for the comments which helped us to improve our manuscript. Detailed answers to all the questions are provided below.

Substantial changes and additions in the manuscript or supplementary materials are marked there in blue text color.

Reviewer #1 (Remarks to the Author):

The revised paper is significantly improved from the previous version that I reviewed for another Nature journal. I have only one remaining suggestion. It seems the current clustering algorithm assumes the same integration site occurs at a single time point. When many integration sites are observed, is it possible that the IS is integrated into indistinguishable, proximal locations in different clones? How this will impact the clustering results based on the abundance patterns of IS insertions? It will be useful to study this impact in the simulation experiments.

We thank the reviewer for this comment. We discussed in the previous round of revisions that, although the presence of identical or indistinguishable IS in two independent clones cannot be fully excluded, this is a negligible problem. The human genome is composed of about 3 billion base pairs, which in principle may all serve as potential IS. Our analysis pipeline focuses on potentially expanding clones and includes a filtering step which in most cases reduces the number of relevant IS to 50-100. Among those, it is extremely unlikely (about one in 10 million) that the same IS is hit twice. Even if a proximity of an IS is considered (which is generally detectable by the slim-PCR approach), such an event remains unlikely. We should point out that the problem of having the same IS in different clones would also occur for conventional IS analysis, in which only IS abundance is monitored without testing for multiple IS occurring in the same clone. We are not aware that this has lately been reported as a problem in gene therapy trials.

While such an approximation, as outlined above, always relies on certain assumptions, we agree with both reviewers 1 and 3 that studying the impact of even this unlikely event is a second step to appreciate the potential error. To this end we set out to assume that one of the IS of the largest clones depicted for the simulations in Figure 3 (different levels of clonal heterogeneity leading to different levels of clonal advantage) is shared with any other IS initialized alongside. Based on this modified time course we can rerun the reconstruction pipeline (repeating the choice of the "sibling IS" 100 times). As we now indicate in the newly added Supplementary Figure S4, we observe no effect on the overall reconstruction quality.

In order to address these concerns we are now explicitly showing the results of the corresponding model simulation in the newly added Supplementary Figure 4 and address the point both in the results part and in our discussion section.

Reviewer #2 (Remarks to the Author):

In my previous comments, I mainly raised the concerns about methods and descriptions. In the revised manuscript, the authors answered these concerns by updating the manuscript and supplementary methods. Overall, it is greatly improved. I have no further concern.

We thank the reviewer for her/his contributions to improve our manuscript.

Reviewer #3 (Remarks to the Author):

I would like to thank the Authors for working on revising their original submission (reviewed in Nature Computational Science), including the revisions made in the repository. I now get a better understanding of the computational challenges of the problem that MultIS aims to solve.

I emphasize that my primary expertise is in Computer Science and Math so my review is focused on these aspects of this work. My main focus is on the following:

- (i) assessing novelty of the proposed approach, in particular with respect to the previously published clustering algorithms.
- (ii) assessing its correctness from the mathematical point of view and its contributions from the computational and mathematical points of view.

After having close look into the methods presentation in the main text, as well as in the supplementary, I have a single major comment/question: what is the main algorithmic novelty of this work? Based on what I could see, it seems to be very minimal in order to justify a publication in a journal Nature Communications. Please note that I am not claiming at all that the package itself is useless and I actually leave a judgement about that to the Reviewers with stronger biological background and more experience in working with this type of data.

We thank the reviewer for this critical appraisal of our work. Indeed, we are not claiming that single elements of our analysis pipeline are algorithmically novel. We were intrigued by an obvious shortcoming that can severely bias the monitoring of artificially marked clones as they are increasingly used in gene therapies and related experimental approaches. As such, we worked on available and simulated data to suggest a tailored approach that particularly addresses this question and can be directly applied to respective data sets. This is something truly missing and represents the novelty of our work.

We would like to point the reviewer to the comments we received from the additional reviewer 4 below, with an expertise in the biological subject who states "*A process or pipeline to identify multiple IS belonging to a single clone would be highly useful for advancing our understanding of clonal dominance and potential for untoward cellular transformation. ... We would consider the manuscript of high value to the field and readership of Nature Communications if these concerns were addressed in a revision.*" We provide detailed answers to the concerns by reviewer 4 below.

I am also slightly concerned that the Authors are having some misunderstanding of certain tools designed for analyzing cancer sequencing data. For example, in the Introduction (Page 3) they mention that PyClone aims to recover the underlying clonal phylogeny, which is not correct (see the cited paper from Nature Methods: Roth et al. 2014). Namely, PyClone does not recover any phylogeny. It just performs clustering based on the read counts (corrected for purity and ploidy). I recommend a careful revision of these statements. Some possible alternatives to "clonal phylogenies" can be "clonal composition" or "clonal structure".

We thank the reviewer for this hint and rephrased the according statement.

In their response to point 3 raised by Reviewer 1, the Authors claim that "the probability of observing two independent events in the same position follows a binomial distribution..." and based on that assumption they obtain extremely low probability for having such events. Similar (but not fully equivalent) conclusions have been repeatedly made and then questioned in the studies of cancer. Therefore I would be a bit more careful with making so explicit statements about which particular distribution we are having in this case. I would rather focus on showing that a couple violations of the assumption do not have a significant impact on the results of the method. I understand that this might be hard to do on the real data, but it should be doable in the simulated setting. Related to this, I also recommend having a short look at the following recent paper <https://www.nature.com/articles/s41588-021-01005-8>.

We thank the reviewer for pointing out this interesting read and agree that our "back of the envelope" estimates might involve some assumptions that are violated in reality. While we are aware that vector insertion is not a completely random process acting on the whole genome, we are only making the point that for the analysis of about 100 larger and repeatedly measurable IS it is very unlikely that they occur in the same position.

We wish to point out that the problem of having the same IS in different clones would similarly occur for conventional IS analysis, in which only IS abundance is monitored without testing for multiple IS occurring in the same clone. We are not aware that this has lately been reported as a problem in gene therapy trials.

We catch up on the suggestion of the reviewer, which was similarly raised by reviewer 1, to use our simulation approach and to investigate to which extent the reconstruction quality is limited by the unlikely event of a "double integration in the same genomic location". To this end we set out to assume that one of the IS of the largest clones depicted for the simulations in Figure 3 (different levels of clonal heterogeneity leading to different levels of clonal advantage) is shared with any other IS initialized alongside. Based on this modified time course we can rerun the reconstruction pipeline (repeating the choice of the "sibling IS" 100 times). As we now indicate in the newly added Supplementary Figure S4, we observe no effect on the overall reconstruction quality.

In order to address these concerns we are now explicitly showing the results of the corresponding model simulation in the newly added Supplementary Figure 4 and address the point both in the results part and in our discussion section.

MINOR:

Now that I better understood the key difference between analyzing tumor and IS data, looking only into Figure 1, I would still be convinced of the applicability of methods for analyzing multi-timepoints tumor sequencing data. However, it might indeed be challenging to come up with a better visualization so I do not necessarily expect any additional updates to this figure, especially considering that there is now some discussion of the tumor data clustering algorithms in the main text.

We agree with the reviewer that this research question still has space for further explorations and we hope that our manuscript can contribute to this process.

Reviewer #4 (Remarks to the Author):

Review: "Clonal reconstruction from co-occurrence of vector integration sites allows accurate quantification of expanding clones in vivo"

Sebastian Wagner, Christoph Baldow, Andrea Calabri, Laura Rudilosso, Pierangela Gallina, Eugenio Montini, Daniela Cesana, Ingmar Glauche

Here Wagner et al. detail a method to quantify the effects of vector copy number when performing viral integration site (IS) analysis in the hematopoietic stem and progenitor cell gene therapy (GT) setting. Multiple studies have demonstrated that two general characteristics of virally-transduced cells can result in increased risk of insertional mutagenesis and subsequent malignant transformation: First, the locus of integration, such as an integration event which activates a proto-oncogene or silences a tumor suppressor gene (i.e. integration in or near a "high risk" genomic locus). Second, the number of individual integration events which occur in a single cell. This is based on the demonstration that multiple integration events may occur in single cells during viral transduction. The higher the number of genomic perturbations, the higher the risk of altered genomic stability. While much of the field has focused on identification of IS associated with high-risk genomic loci, the ability to identify cells with multiple IS events has received far less attention. Thus, there is a clear need to develop better methods for estimating clonal dominance even when masked by high vector copy numbers per cell.

In current analysis pipelines, the genomic site of (semi-random) provirus integration is used as a cellular barcode to track clones after transplant. However, this approach treats each identified IS as a unique clone because the present sequencing methodology does not include cellular identifiers which can attribute IS to the cell from which they came. In the case where multiple integration events occur in one cell, these IS will be inaccurately called unique clones in followup procedures. This could lead to an underestimation of clonal dominance and, importantly, could result in delayed identification of clones with transformative potential.

A process or pipeline to identify multiple IS belonging to a single clone would be highly useful for advancing our understanding of clonal dominance and potential for untoward cellular transformation. Here the authors focus on a pipeline analysis which could be applied to existing IS data for identification of unique IS with high probability of being present in the

same cell. This analysis is dependent on the method of IS analysis capturing an accurate representation of clonal contribution over time and, the largest clonal pools differing enough in size and/or differential contribution to blood lineages in hematopoiesis. Under these conditions, a clustering algorithm could be used to group IS that are detected similar in quantity and lineage contribution over time into units that represent individual clones. While this methodology is timely and has great potential, the presented experiments could be greatly strengthened by the items mentioned below. We would consider the manuscript of high value to the field and readership of Nature Communications if these concerns were addressed in a revision.

We thank the reviewer for this positive appraisal of our work. We are convinced that addressing mentioned items further improved our manuscript. Detailed answers are provided below.

Major Revisions Requested:

1. Detection of aberrant hematopoiesis in GT setting at the earliest possible time is clearly important. The ability to predict aberrant hematopoiesis before clinical emergence is the holy grail in GT, but to do this requires careful study of existing cases of insertional mutagenesis as a result of multiple IS per cell. A data set of clone tracking in a GT setting in hematopoietic stem and progenitor cells where a single clone with multiple IS became dominant exists¹. It would strengthen this paper if this algorithm could be run on this existing dataset and determine the earliest point where the multiple IS from the dominant clone could be identified as a single clone. Alignment of this detection with the clinical data from this subject would help determine how much earlier risk could have been identified. This would serve as a real-life proof-of-concept for the utility of the presented algorithm.

We thank the reviewers for pointing us to this data set. Applying our method to this data set impressively underlines the applicability of our approach. Our approach correctly identified the six IS within the time course data that jointly made up for the dominating clone (new Figure 7). The additional three IS identified with the amended primer design cannot be reconstructed if only time course data is considered. We also demonstrate that the (almost) correct association of IS can be inferred much earlier (new Supplementary Figure S10). The Figure below (also added as Supplementary Figure S11) demonstrates that the correct assignment of the six relevant IS to the same clones could have already been detected at time point 187 days.

Within our manuscript, we added a description of these findings to our Results part within an additional section named “Clonal reconstruction for a non-human primate data set”. We also added Figure 7, Supplementary Figures S10 and S11. Further minor adaptations were necessary throughout the manuscript to account for this additional data set. A brief description of the experimental setting is also added to the Supplementary Materials, paragraph “IS measurements from a study on rhesus macaques”.

2. The above algorithm check in the setting of a documented, multiple IS clone which displayed insertional mutagenesis is one proof-of-concept. Key to demonstrating a useful analysis pipeline is estimating how often the analysis incorrectly identifies multiple-IS clones (false positive call rate) and misses relevant multiple-IS clones (false negative call rate). The authors could demonstrate in an experimental setting with PCR confirmation of individual clones, the frequency of correct multiple IS calls in a given dataset. This could be done with hematopoietic colony-forming cells in an *in vitro* setting, which takes just a few weeks to generate, and could be compared to VCN from the same dataset of individual colonies.

We thank the reviewer for the suggestion. There are two aspects we would like to touch before explaining the changes we did. First, a false positive (FP) or a false negative (FN) rate is most suited if only one clone is considered to which a given IS either belongs or not. Assessing multiple clones at once the question might be rephrased as to address how well the overall clustering fits with the given ground truth. This multi-clonal view is much better reflected with the adjusted Rand Index (ARI) that we are using throughout the manuscript to measure the quality of our reconstruction process. Second, the reviewer is correct that a culture setting with only a few clones with known IS is an ideal setting to verify our approach. We would like to draw the attention to the *in vitro* data set already reported in our manuscript (Results section “*In vitro* validation assays confirm the validity of clonal reconstruction.”), in which we use transduced and isolated K562 clones that were mixed in different ratios. We then performed IS retrieval and confirmed all IS by custom PCR, in line with the Reviewer’s comment, whose IS are reported in Supplementary Table S2. On these IS and the different

mixes we could validate and benchmark our method. Instead of using FP and FR, we measured precision (also referred to as positive predictive value ($TP/(TP+FP)$)) and recall (also referred to as sensitivity ($TP/(TP+FN)$)). We evaluated these values independently for the three clones ID#30, ID#37 and ID#46 containing 4, 6 and 10 IS, respectively. Depending on the number of mixes used for the reconstruction (ranging from 3 to the maximum number 7), we observe consistently high values for both precision and recall (see figure below which is also added as a new subfigure to Figure 5), especially if at least five different mixes are considered. These results confirm that our method correctly identifies different IS belonging to the same clones and benefits from adding more mixtures/measurements.

In order to address this important point within our manuscript we adapted the description in the section on “In vitro validation assays confirm the validity of clonal reconstruction.” and now explicitly refer to precision and recall for those three relevant clones. We also amended Figure 5 to include the new subfigure.

3. The need to understand the effects of multiple IS in individual clones is clearly valid and clustering IS based on similarities of quantity and lineage contribution seems correct. I think the validation on mouse and fixed populations were good. In the mouse experiment were any of the VCN numbers from simulation verified? As noted in point 2, connection of VCN and algorithm-called multiple-IS clones seems like an important check to perform, at least in an in vitro setting. Another option would be to use publicly accessible data from a study where VCN on single colonies was performed and run the algorithm on this dataset.

We thank the reviewer for this comment and fully agree that a correspondence between the number of reconstructed IS per clone and the experimentally confirmed VCN further strengthens our reasoning. For the validation assay based on the generated mixes of four clones with different numbers of IS, our suggested reconstruction process recovers the known ground truth (see above). The same is true for the newly included macaque data set, in which the number of repeatedly detectable IS is perfectly met (new Figure 7). For the mouse data, we are indeed left with comparing the number of IS per clone reconstructed with our *MultIS* pipeline to the experimentally determined VCN. This latter value has been

measured by quantitative droplet digital PCR (ddPCR) for most assays and correlates well with the average number of IS obtained by *MultIS*. We need to point out that this latter estimate is only obtained by analyzing a subset of dominating IS and might therefore diverge from the experimental techniques based on bulk analysis.

We now show this data in the newly added Supplementary Table S5, which is also referenced in the Materials and Methods section.

4. The assumptions made about a clone population in a non-human primate (NHP) or human setting are concerning. The algorithm for predicting HSC clonal dynamics predicts the generation of pools of HSC clones that happen to replicate and not differentiate. For a low number of clones, (e.g. Figure 3), using parameters of this model that assume inheritable differences in clone growth rate and clone lineage bias makes clones easily distinguishable. However, both assumptions may be optimistic for the case of clone identification. For the case of an NHP or human transplant, the initial number of clones is likely to be 100,000-1,000,000 or more. Published hematopoietic engraftment and contribution dynamics in NHP and humans suggest this HSC diversity will drop rapidly to 1,000-10,000 unique clones, each represented by a large pool of HSCs. Clustering will in part depend on differences in clone pool size, but many of the top pools will have very similar size, especially if no parameter is added to give some clones a growth advantage. The idea that clones have a built-in bias of the lineages they produce is also optimistic. To date, IS processing has not been done to a level that proves clonal bias. If these assumptions are taken out of the “ground truth” model and a more realistic clone population is simulated, would the clustering still be accurate?

Two simulations with identical starting condition that are close to an actual GT application would be helpful. In one case no clone has a growth advantage. Using this case, are IS incorrectly clustered

together? In the next case if one high VCN clone is given a growth advantage, are these IS identified as one clone and at what time point does this become apparent?

We thank the reviewer for bringing up those questions. We would like to clarify that within the model all clones replicate AND differentiate. We are only assuming the differentiation rates mildly differ, giving some clones a clonal advantage over others. This way, we can tune how fast a simulated system will change towards oligo- or monoclonality (compare Figure 3). For illustration, the model simulations for Figure 3 have been initialized with 100 clones each holding 100 cells initially. However, we verified that starting the simulations with more but smaller clones does not at all change our reconstruction quality (see newly added Supplementary Figure S3). This results (i) from the rather rapid drop of HSC diversity (which is also observable in the model) and (ii) from focusing the reconstruction process on the most abundant IS only (“filtering”) at the final time point of the analysis.

We further agree with the reviewer that the true level of clonal bias has not been studied in depth while this is a very interesting question. However, there are published findings supporting our assumption to which we also refer in our manuscript. Furthermore, Figure 6 in our manuscript shows that clone 1 (shown in red) is prominent among MNC, CD3 and CD19 cells but rarely present among CD11b cells. Similar phenomena are observable for all the other mouse data sets reported in Supplementary Figures S6 to S9. The same is also true

for the added data in Figure 7 (macaque data). Here, the “red” clone dominates granulocytes and monocytes but is less abundant among B and T cells. We wish to point out that while our reconstruction pipeline benefits from such a clonal bias, it still works without: all simulations in Figure 3 study precisely this scenario without assuming any clonal bias.

Finally we take up the suggestion of the reviewer to compare two extreme scenarios (“equal clones” vs. “strong clonal advantage”). This setting is the principle idea behind Figure 3. While the left column refers to the case that all clones are equal (please note that there is still a slow conversion given the intrinsic stochasticity of replication and differentiation), the clonal differences increase towards the right columns. In the rightmost column, one clone is randomly receiving the highest clonal advantage (i.e. lowest differentiation rate) and progressively outcompetes all other clones. The same can be achieved if we take the leftmost scenario (“all clones are equal”) and artificially increase the competitive advantage of one of the clones. The set of figures below represents an additional simulation for this scenario for the case that the first (topmost) clone is favored. This scenario is virtually identical to the rightmost scenario in the current Figure 3. The lower panel in Figure 3 (subfigures j,k and l) indicates that a clonal reconstruction is feasible with good quality for each of those settings as long as a sufficient number of measurements is available.

The reviewer also raises the question at which time point the IS are correctly identified as one clone. Supplementary Figure S11 hints towards the fact that an early identification is possible, which is further supported by our findings reported in Figures 3 and 4 outlining that with about 8 independent measurements rather good reconstructions can be achieved. However, we feel that a systematic assessment of this question is beyond the scope of this work and would require a different perspective in which the cumulative assessment of temporal measurements is more in focus rather than the proof of concept that the identification of different IS in the same clone is generally feasible.

In order to adhere to the suggestions of the reviewer and to more clearly outline our motivation behind Figure 3 we amended our descriptions at different points. In particular, we make clear that increasing the heterogeneity parameter δ in our simulation model reflects the suggested scenarios from “equal clones” towards “pronounced clonal advantage”. We also include a comment on the assumptions regarding the clonal bias in the newly added section on the macaque data. We furthermore added Supplementary Figure S3 to illustrate that initializing the system with more but smaller clones does not change our reconstruction quality.

5. This process is dependent on the reliable capture of IS over time. In the introduction section, a value of 0.1% is given for the reliable quantification of IS abundance. However, the threshold for reliable quantification of IS abundance will depend on gene marking level of the test organism, and the methodology applied to identify IS, which can vary from group to group. At minimum, we would like to see a review of the available study data and an estimated range of this value based on reported data, with simulations that cover at least the two extremes and mean of this range.

For example, in an autologous transplanted NHP, gene marking levels in peripheral blood white blood cells may be 30% or higher and if a fluorescent reporter gene is included in the viral vector, flow sorting could be used to purify IS containing cells. This means in a typical sample (500,000 genomes), each IS at 0.1% would be present at 500 copies. IS capture efficiency has been demonstrated to be low (~5%).² This would result in an average of about 25 good IS sequence reads. If the gene marking level is low, such as ~5% as has been demonstrated in the human GT setting, then a 0.1% clone would be present in $500,000 * 0.05 * 0.001 = 25$ copies. If IS capture is about 5% then this would mean an average of 1.25 sequence reads for this clone. This is concerning for reliable detection. For groups who want to apply this algorithm, what are the parameters of gene marking and IS quantification accuracy that will generate a reliable result?

The reviewer pointed out an important aspect, that is providing estimates for users wishing to apply the *MultIS* pipeline in an experimental setting. We agree with the reviewer that a reliable quantification of IS is crucial for the successful application of the reconstruction pipeline and that this quantification depends on several factors. However, we wish to point out that the detection limit itself is not the major limiting factor of our approach as long as the analysis focuses on those more abundant IS that can reliably be quantified. This is what our suggested filtering function provides.

We follow the argument of the reviewer and agree that low levels of gene marking increase the sampling error. This limitation does not only account to the application of our method but to any assessment of IS dynamics even without considering the true clonal structure. However, this limitation can only be counteracted by increasing the sample size to obtain enough DNA material harboring IS, while the clonal reconstruction pipeline only starts thereafter. This is also the reason why we chose to work with percentages to overcome any issues related to the amount of genomic material and in general the number of cells since this is not the focus of this work.

On the technical side, we are aware that genetic barcodes (which are not applicable in a therapeutic setting) or sonication based methods most likely provide the best results with the smallest quantification error. Indeed, a valid suggestion to end users could be to quantify the amount of genomic material such that any potentially expanding clone could be visible, both in a polyclonal or oligoclonal sample. This is the reason why we reported the detection limit of the SLiM-PCR (set at 0.1%) and we selected the minimum threshold for clonal abundance at 1%. Independently of the absolute number of genomic fragments per IS, if a clone is detected above the threshold it means that all the IS belonging to that clone will have a similar abundance (in percentage), with some exceptions depending on alignments or mappability of some regions etc. Any abundance detected below 0.1% for a single IS cannot

be considered highly reliable (variability > 20% and signal-to-noise ratio above thresholds). Indeed, if a clone with multiple IS is captured in one sample, our working assumption is that the abundance of all those IS is correlated, and if the same clone is recaptured several times in independent samples, for each sample the relative abundance of its IS will remain comparable. For this reason, we argue that any sample processed by IS retrieval and passing minimum data quality filters (number of reads, number of genomes and quantification, etc.) will be useful to extract the internal correlation among the IS belonging to a single observed clone.

We explicitly use our model simulations to address the dependency of the reconstruction quality on the quantification error. Figure 4 of the current manuscript studies different extreme cases, in which increasing levels of technical/measurement noise are superimposed on the particular clonal simulation data. We show that with a standard deviation in the order of 4 to 5 % ($\sigma = 0.04$ to 0.05) appears unproblematic (this corresponds to a situation in which 95% of repeated measurements are within a +/- 8-10% range of the true value), while even higher levels can be handled if there is sufficient heterogeneity in clonal contribution for different hematopoietic subcompartments.

In order to address the concerns of the reviewer, we touch on those technical limitations and provide a brief review of the relevant study data in the introduction section. Furthermore, we explicitly highlight our analysis of the measurement/technical noise in Figure 4. We also provide an estimation of how the simulated measurement noise (in terms of the parameter σ translates into a confidence region for multiple quantifications of the same IS). A more practical interpretation has also been added to the discussion section.

For clones detected, whatever their VCN state, at what level of contribution do they become concerning?

We thank the reviewer for this question, which we explicitly separated from the above block of questions. Indeed, this is the crucial question, which we cannot answer, not within the scope of this manuscript, but also not in general. Regulatory authorities usually impose fixed threshold, although we think that many aspects need to be considered, such as the overall level of clonal marking and the average VCN. Moreover, we would argue that it is in particular the “dynamics” (meaning the change over time) that actually reveals the clonal potential. We see our work as an important contribution to recover the “clonal dynamics” rather than to stay with the IS abundance.

Reviewers' Comments:

Reviewer #1:

Remarks to the Author:

The authors have addressed my comments satisfactorily.

Reviewer #3:

Remarks to the Author:

I thank the Authors for responding to my comments. Based on their responses and the feedback received from the other reviewers, including the reviewer more closely familiar with the direct applications of this pipeline in analyzing real data, I have no further comments.

Reviewer #4:

Remarks to the Author:

The authors have done a very nice job with the revision and have made significant improvements to the manuscript and study with added figures and analyses. The rebuttal was exceptionally well-addressed and we thank them for the careful thought and thorough effort. We also appreciate their enthusiasm in experimental validation of their approach, it has greatly strengthened the paper, which we believe is acceptable for publication and will be of high impact. We believe this tool will move the field forward and reveal new limitations to be addressed in future studies. We appreciate the opportunity to review this work.

Sincerely,

Drs. Jennifer Adair and Mark Enstrom